# TMA from Cosines of Conical Angles Acquired by a Towed Array

**DOI:** 10.3390/s21144797

**Published:** 2021-07-14

**Authors:** Antoine Lebon, Annie-Claude Perez, Claude Jauffret, Dann Laneuville

**Affiliations:** 1Naval Group, 83190 Ollioules, France; antoine-lebon@etud.univ-tln.fr; 2Université de Toulon, Aix Marseille Univ, CNRS, IM2NP, CS 60584, 83041 Toulon, CEDEX 9, France; annie-claude.perez@univ-tln.fr; 3Naval Group—Technocampus Océan, 44340 Bouguenais, France; dann.laneuville@naval-group.com

**Keywords:** target motion analysis, observability, fisher information matrix, Cramér–Rao lower bound, conical angles, nonlinear estimation

## Abstract

This paper deals with the estimation of the trajectory of a target in constant velocity motion at an unknown constant depth, from measurements of conical angles supplied by a linear array. Sound emitted by the target does not necessarily navigate along a direct path toward the antenna, but can bounce off the sea bottom and/or off the surface. Observability is thoroughly analyzed to identify the ghost targets before proposing an efficient way to estimate the trajectory of the target of interest and of the ghost targets when they exist.

## 1. Introduction

Bearings-only target motion analysis (BOTMA) is a problem that has been widely studied and various solutions have been proposed in the literature: batch [1,2,3,4,5] or recursive filter (such as extended Kalman filter [6,7,8], unscented Kalman filter [9], particle filter [10], modified instrumental variable [11,12,13]), or a mix of recursive and batch methods [14]. Citing all the papers dealing with this topic is now a hard task. Among the abundant literature, most papers share the same assumption: the target is moving in a straight line with a constant speed, while the passive observer is maneuvering adequately in order to ensure the observability of the target [15,16,17]. The bearings are the measurements.

In this paper, we are concerned with the same problem, except that the available measurements are the cosine of the relative bearings, also called conical angles because the target belongs to the cone of ambiguity whose revolution axis is the line along which the towed array is moving (see [18] p. 39). Implicitly, we consider a target moving in 3D at a constant and unknown depth in near field; in this case, the two more energetic rays are the direct and the reflected paths (bottom or surface). In most cases, the sound bounces off the sea bottom. Therefore, we extend our analysis to surface- and sea bottom-bounced rays.

Indeed, the array detects the cosine of the relative angle of the direction of arrival by means of a suitable spatial filtering method such as beamforming, or more sophisticated techniques (see [19]). In the near field, sound can propagate to the sensor array along the direct path and/or the bottom-reflected path, and/or the surface-reflected path. Most of the time, at most, two rays coming from the same target are detected [18,20].

Unlike Gong [21] and Blanc-Benon [22], who addressed the three-dimensional target motion analysis (TMA) from a sequence of time differences of arrival (TDOA) of a signal traveling by two different paths coupled with a sequence of azimuths, we assume in this paper that the available measurements are the cosines of the conical angles only. In [23], a similar problem was addressed, but observability was not studied. We will consider two situations: the first case is devoted to TMA when sound propagates along a non-direct path at each sampling time. This will be the topic of Section 3: we will conduct observability analysis and identify all the ghost targets, given a set of noise-free measurements. We will prove that an assumption of the target’s depth makes the target’s trajectory observable, but not estimable (in the sense that the asymptotic performance given by the Cramér–Rao lower bound—CRLB—of the estimator of the depth is out of the physical constraints, that is, the source is navigating between the surface and the sea bottom).

In the fourth section, we will consider scenarios in which the antenna changes its own route. We will prove that the trajectory of the target is almost certainly observable.

In the fifth section, we will assume that sound will propagate along the direct path and the bottom-reflected path. The two rays will be assumed as being detected. Observability analysis will reveal that only three ghost targets at most exist without maneuvering the antenna. We will check that, in this case, the depth is not “estimable”. We will give a palliative, allowing us to propose an estimator which is operationally acceptable, the price being a small bias. Convincing simulations will be given at the end of this section, proving that, even when the duration of the scenario is short, the estimated trajectory is very close to the true one. A conclusion ends the paper.

## 2. Notation and Problem Formulation

We consider two underwater vehicles moving at their own constant depth. The first mobile is a surface vessel or a submarine towing a horizontal sensor array, and the second one is the target of interest. Given a Cartesian coordinate system, the acoustic center of the array is located at time *t* at ${\left(\begin{array}{ccc} x_{O}\left(t\right) & y_{O}\left(t\right) & z_{O} \end{array}\right)}^{T}$. At the same time, the target is at ${\left(\begin{array}{ccc} x_{T}\left(t\right) & y_{T}\left(t\right) & z_{T} \end{array}\right)}^{T}$. The respective horizontal positions of the target of interest and of the center of the array at time *t* are denoted by $P_{T}\left(t\right)={\left(\begin{array}{cc} x_{T}\left(t\right) & y_{T}\left(t\right) \end{array}\right)}^{T}$ and $P_{O}\left(t\right)={\left(\begin{array}{cc} x_{O}\left(t\right) & y_{O}\left(t\right) \end{array}\right)}^{T}$. The sea bottom depth (assumed to be a constant) is denoted as *D*. The source is said to be endfire to the line array if its trajectory is in the same line as the array (which implies that the array and the source are at the same depth, and share the same route). It is broadside to the antenna if it navigates in the vertical plane orthogonal to the line array and passing by the acoustic center of the array. The sensor array detects the line of sight of the target; more precisely, ad hoc array processing (or spatial filtering) delivers at time *t* the cosine of the conical angle $c_{a}\left(t\right)$ given by $\cos\left(c_{a}\left(t\right)\right)=\cos\left(\theta \left(t\right)-h\left(t\right)\right)\cos\left(\phi \left(t\right)\right)≜m\left(t\right)$, where θ(t) and ϕ(t) are, respectively, the azimuth (or bearing) and the elevation of the path along which the sound emitted by the source propagates. The angle h(t) is the heading of the sensor array. Denoting the relative position coordinates of the source with reference to the acoustic center of the array by $x_{OT}\left(t\right)=x_{T}\left(t\right)-x_{O}\left(t\right)$ and $y_{OT}\left(t\right)=y_{T}\left(t\right)-y_{O}\left(t\right)$, we have $\theta \left(t\right)=arctan\left(x_{OT}\left(t\right),y_{OT}\left(t\right)\right)$. Figure 1 displays the different angles and the two actors (the observer reduced to the linear array, and the target).

The ray of the sound (or signal) emitted by the source can be reflected by the bottom and/or the surface or travels in the surface or deep channel. The sound–speed profile makes the paths curve. In this paper, we will consider that the target is in the near field (the distance between the source and the array is less than 20 km), and the bottom depth is in the range 2000–5000 m. Due to the large curvature of the ray (about 80 km), we will approximate the path of the sound as a set of zigzags defined by the reflections on the bottom or on the surface. So, we implicitly use the Snell law widely employed in geometrical optics. An image-source is created whose depth $\zeta_{T}$ will be called “image-depth”. A path is then defined by the triplet $\left(\delta ,n_{B},n_{S}\right)$, where
δ indicates the direction of the path of the sound emitted by the source: if the path is toward the surface, δ=−1, otherwise δ=+1,$n_{B}$ is the number of bottom reflections, and$n_{S}$ is the number of surface reflections.

Figure 2 illustrates three different paths

We have to consider the depth difference between the array and the image-source defined by $\zeta_{OT}≜\zeta_{T}-z_{O}$ if the ray has been reflected (by the sea bottom or by the surface), or $\zeta_{OT}≜z_{T}-z_{O}$ if the sound wave uses the direct path.

A general expression of $\zeta_{OT}$ based on the triplet $\left(\delta ,n_{B},n_{S}\right)$ is given by $\zeta_{OT}\left(\delta ,n_{B},n_{S}\right)=-2\delta n_{B}{\left(-1\right)}^{n_{S}+n_{B}}D-z_{O}+{\left(-1\right)}^{n_{S}+n_{B}}z_{T}$. Note that, given the path, the link between $\zeta_{OT}\left(\delta ,n_{B},n_{S}\right)$ and $z_{T}$ is linear: $\zeta_{OT}\left(\delta ,n_{B},n_{S}\right)=az_{T}+b$, the constants being a function of the triplet $\left(\delta ,n_{B},n_{S}\right)$, D, and $z_{O}$. Moreover, $\zeta_{OT}\left(\delta ,n_{B},n_{S}\right)$ is null if and only if the antenna and the target are navigating at the same depth ($z_{T}=z_{O}$), and sound is traveling in the direct path. In this case, cos(ϕ(t))=1. For the sake of simplicity of the notations, we will simply subsequently denote $\zeta_{OT}$ instead of $\zeta_{OT}\left(\delta ,n_{B},n_{S}\right)$.

For the above examples, we have $\zeta_{OT}\left(1,0,0\right)=z_{T}-z_{O}$ (direct path), $\zeta_{OT}\left(+1,1,0\right)=2D-\left(z_{T}+z_{O}\right)$ (bottom-reflected path), and $\zeta_{OT}\left(+1,2,1\right)=4D-\left(z_{T}+z_{O}\right)$ (bottom-surface-bottom reflected path). Note that $\zeta_{OT}\left(\delta ,n_{B},n_{S}\right)$ can be negative (the image-source is above the surface). Consequently, the cosine of the elevation is $\cos\left(\phi \left(t\right)\right)=\frac{\sqrt{x_{OT}^{2}\left(t\right)+y_{OT}^{2}\left(t\right)}}{\sqrt{x_{OT}^{2}\left(t\right)+y_{OT}^{2}\left(t\right)+\zeta_{OT}^{2}\left(\delta ,n_{B},n_{S}\right)}}$.

Figure 3a displays the cone of ambiguity, defined by the set of sources sharing the same cos(ϕ(t)). In Figure 3b, we plot a direct ray and a bottom-bounced ray, which allows us to figure out the various angles with which we will work.

We assume that the source is moving in constant velocity (CV) motion during the scenario. Our challenge is to estimate its trajectory, i.e., the state vector defining it, $X≜{\left(\begin{array}{ccccc} x_{T}\left(t^{*}\right) & y_{T}\left(t^{*}\right) & z_{T} & {\dot{x}}_{T} & {\dot{y}}_{T} \end{array}\right)}^{T}$, for a chosen $t^{*}$, from noisy measurements.

We consider two situations:
Only one ray is detected by the array during the scenario; in this case, we have at each time *t* a measurement m(t), given the path along which the wave propagates.Two rays (traveling on two different paths) arrive at the sensor’s antenna. In this case, the available measurement at time *t* is a couple of measurements, say $\left(m_{1}\left(t\right),m_{2}\left(t\right)\right)$, given the two paths along which the wave propagates.

After the spatial filtering, the antenna supplies a noisy measurement of m(t) or a noisy measurement of $\left(m_{1}\left(t\right),m_{2}\left(t\right)\right)$. The noisy measurements are regularly acquired at $t_{k}=\left(k-1\right)\Delta t$, k∈{1,…,N}, for a fixed sampling time Δt.

Before attempting to estimate X, we must answer several questions:
Is the vector X observable from the set of measurements {m(t),t∈[0,T]}? Note that, in TMA problems, observability is often analyzed in continuous time (see [15,17], for example), even though the noisy measurements are given in discrete time.If not, what are the ghost targets (those which could be detected at the same set of measurements {m(t),t∈[0,T]})?How do we make X observable or with which new information?Is the vector X observable from the set of couples $\left\{\left(m_{1}\left(t\right),m_{2}\left(t\right)\right),t\in \left[0,T\right]\right\}$?

For the cases where X is observable, we have then to compute the asymptotical performance of an unbiased estimator (given by the CRLB [24]), and the performance of our estimators in terms of bias and the covariance matrix. It is worth noting that using the FIM to prove observability can lead to a wrong conclusion [25]. This why we use an analytic approach.

## 3. TMA from One Ray

In this section, we consider the case where the array collects the cosine of a conical angle, the path of the ray being known by the operator. We start by analyzing the observability of the trajectory of the source of interest.

### 3.1. Observability Analysis

**Theorem** **1.**
*Let a linear antenna measure the cosine of a conical angle in the direction of a source, both in CV motion. The path of the sound emitted by the source is known, as is also the sea bottom depth.*
*1.* 
*If the target is broadside to the antenna, then the set of ghost targets is composed of virtual sources broadside to the antenna.*
*2.* 
*If the target is endfire to the antenna, the set of ghost targets is composed of virtual sources endfire to the antenna.*
*3.* 
*If the target has the same heading as the array (but is not endfire to it), then the set of ghost targets is composed of virtual targets with the same heading as the antenna. More precisely, the ghost image of each ghost target is moving on a cylinder whose axis is the antenna axis, and whose radius is a positive scalar β. The relative ghost target velocity is equal to β times the target’s velocity. The initial distance between the ghost image and the center of the antenna is equal to β times the initial distance between the target-image and the center of the antenna.*
*4.* 
*In any other case, for a chosen image-depth $\zeta_{G}$, the set of ghost targets is composed of virtual targets whose motion relative to the array is defined by $P_{OG}\left(t\right)=\beta P_{OT}\left(t\right)$ or $P_{OG}\left(t\right)=\beta SP_{OT}\left(t\right)$, where S is the 2D axial symmetry around the line of the array, and β is a positive scalar. The scalar β is equal to $\frac{\left|\zeta_{OG}\right|}{\left|\zeta_{OT}\right|}$ if $\zeta_{OT}\neq 0$. If $\zeta_{OT}=0$ (which can happen with a direct path only), β can have any positive value.*



Preamble: In the following proof, we choose $t^{*}=0$. Instead of working with the state vector $X={\left(\begin{array}{ccccc} x_{T}\left(0\right) & y_{T}\left(0\right) & z_{T} & {\dot{x}}_{T} & {\dot{y}}_{T} \end{array}\right)}^{T}$, we will use the relative state vector of the image source, which is $Y≜{\left(\begin{array}{ccccc} x_{0T}\left(0\right) & y_{OT}\left(0\right) & \zeta_{OT} & {\dot{x}}_{OT} & {\dot{y}}_{OT} \end{array}\right)}^{T}$. The reason for this is that we are able to recover X from Y without ambiguity.

We will prove this theorem in the special case where the heading of the antenna is equal to 0°, and the value $y_{OT}\left(t\right)$ is positive. This can be easily obtained with an ad hoc rotation of the whole scenario. This will simplify the expression of the measurement, without loss of generality.

**Proof of Theorem 1.** We are seeking the ghost target whose horizontal position at time *t* is ${\left(\begin{array}{cc} x_{G}\left(t\right) & y_{G}\left(t\right) \end{array}\right)}^{T}$, detected in the same cosine of the conical angle, that is$\frac{y_{OT}\left(t\right)}{\sqrt{x_{OT}^{2}\left(t\right)+y_{OT}^{2}\left(t\right)+\zeta_{OT}^{2}}}=\frac{y_{OG}\left(t\right)}{\sqrt{x_{OG}^{2}\left(t\right)+y_{OG}^{2}\left(t\right)+\zeta_{OG}^{2}}}$, with $x_{OG}\left(t\right)=x_{G}\left(t\right)-x_{O}\left(t\right)$, $y_{OG}\left(t\right)=y_{G}\left(t\right)-y_{O}\left(t\right)$, and $\zeta_{OG}$ is the image-depth of the ghost target. This equality is equivalent to
(1)$$\frac{y_{OT}^{2}\left(t\right)}{x_{OT}^{2}\left(t\right)+y_{OT}^{2}\left(t\right)+\zeta_{OT}^{2}}=\frac{y_{OG}^{2}\left(t\right)}{x_{OG}^{2}\left(t\right)+y_{OG}^{2}\left(t\right)+\zeta_{OG}^{2}}$$
Note that because the target is moving (as is the ghost target also), the denominators of the left term and of the right term of (1) are two polynomial functions of degree 2.Case 1: $y_{OT}\left(t\right)$ is a zero function, i.e., $\forall t\;y_{OT}\left(t\right)=0$.This means the source is broadside to the antenna: $Y_{T}={\left(\begin{array}{ccccc} x_{0T}\left(0\right) & 0 & \zeta_{OT} & {\dot{x}}_{OT} & 0 \end{array}\right)}^{T}$.In this case, $y_{OT}\left(t\right)=0$, ∀t∈[0,T]. Hence, the set of ghost targets is composed of the virtual targets broadside to the antenna: $Y_{G}={\left(\begin{array}{ccccc} x_{0G}\left(0\right) & 0 & \zeta_{OG} & {\dot{x}}_{OG} & 0 \end{array}\right)}^{T}$.Case 2: $y_{OT}\left(t\right)$ is not a zero function.If ${\dot{y}}_{OT}=0$, then $y_{OT}\left(t\right)$ is a constant. To respect the degrees of the terms of (1), $y_{OG}\left(t\right)$ is a constant too.If ${\dot{y}}_{OT}\neq 0$, then there is a root, say $\tilde{t}$, such as $y_{OT}\left(\tilde{t}\right)=0$, since $y_{OT}\left(t\right)$ is a polynomial function of degree 1. Consequently, $y_{OG}\left(\tilde{t}\right)=0$, and $\forall t\neq \tilde{t}$, $y_{OG}\left(t\right)\neq 0$.We deduce that, in both cases (${\dot{y}}_{OT}=0$, and ${\dot{y}}_{OT}\neq 0$), there exists a positive value β such that $y_{OG}\left(t\right)=\beta y_{OT}\left(t\right)$.
(2)$$\left(1\right)\Leftrightarrow \left\{\left[x_{OG}^{2}\left(t\right)+y_{OG}^{2}\left(t\right)+\zeta_{OG}^{2}\right]-\beta^{2}\left[x_{OT}^{2}\left(t\right)+y_{OT}^{2}\left(t\right)+\zeta_{OT}^{2}\right]\right\}y_{OT}^{2}\left(t\right)=0\;\Leftrightarrow x_{OG}^{2}\left(t\right)+\zeta_{OG}^{2}=\beta^{2}\left[x_{OT}^{2}\left(t\right)+\zeta_{OT}^{2}\right]$$
The quantity $x_{OT}^{2}\left(t\right)+\zeta_{OT}^{2}$ can be equal to zero at any time, or at one time or never.Subcase 1: ∀t, $x_{OT}^{2}\left(t\right)+\zeta_{OT}^{2}=0$.Then, $x_{OT}\left(t\right)=0$, ∀t and $\zeta_{OT}=0$. Note that this case is the one when the target is traveling in the endfire to the array and at the same depth as the antenna and the path is the direct one. For the same reason, $\;x_{OG}\left(t\right)=0$, ∀t and $\zeta_{OG}=0$. The set of ghost targets is hence composed of virtual targets traveling in the endfire to the array and at the same depth as the antenna.Subcase 2: $\exists \overset{˘}{t}$ such that $x_{OT}^{2}\left(\overset{˘}{t}\right)+\zeta_{OT}^{2}\neq 0$.We deduce from (2) that
(3)$$x_{OG}^{2}\left(0\right)=\beta^{2}x_{OT}^{2}\left(0\right)+\beta^{2}\zeta_{OT}^{2}-\zeta_{OG}^{2}$$
(4)$$x_{OG}\left(0\right){\dot{x}}_{OG}=\beta^{2}x_{OT}\left(0\right){\dot{x}}_{OT}$$
(5)$${\dot{x}}_{OG}^{2}=\beta^{2}{\dot{x}}_{OT}^{2}$$
If ${\dot{x}}_{OT}=0$, then$Y_{G}={\left(\begin{array}{ccccc} \pm \sqrt{\beta^{2}x_{OT}^{2}\left(0\right)+\beta^{2}\zeta_{OT}^{2}-\zeta_{OG}^{2}} & \beta y_{OT}\left(0\right) & \zeta_{OG} & 0 & \beta {\dot{y}}_{OT} \end{array}\right)}^{T}$, for any positive constant β and any positive constant $\zeta_{OG}$ less than $\sqrt{\beta^{2}x_{OT}^{2}\left(0\right)+\beta^{2}\zeta_{OT}^{2}}$. Note that, when ${\dot{y}}_{OT}=0$, the target is motionless relative to the center of the array (both have the same velocity); and when ${\dot{y}}_{OT}\neq 0$, the target has the same heading as the array.If ${\dot{x}}_{OT}\neq 0$, then squaring the elements of (4), and using (5), we draw from (3) that $\beta^{2}\zeta_{OT}^{2}=\zeta_{OG}^{2}$. If $\zeta_{OT}=0$, then $\zeta_{OG}=0$, and the scalar β can take any positive value; else $\beta =\frac{\left|\zeta_{OG}\right|}{\left|\zeta_{OT}\right|}$. In both cases, the trajectory of a ghost target is defined by the state vector $Y_{G}={\left(\begin{array}{ccccc} \pm \beta x_{OT}\left(0\right) & \beta y_{OT}\left(0\right) & \beta \zeta_{OT} & \pm \beta {\dot{x}}_{OT} & \beta {\dot{y}}_{OT} \end{array}\right)}^{T}$. □

**Remark** **1.**
*1.* 
*When the source and the observer are at the same depth, and the path is direct, Theorem 1 recovers the conclusions given in [26].*
*2.* 
*The cases (1), (2) and (3) of Theorem 1 are “rare events”, since the events of dealing with a source in endfire, broadside or having the same heading as the antenna during the scenario occur with a probability equal to 0.*
*However,*
*when the target has a trajectory close to one of these special cases, the estimates will have a poor behavior.*
*3.* 
*For case (4), when the detected ray is not a direct path, for example, when the ray is bottom-reflected, a hypothesis about the source is sufficient to obtain one solution, corresponding to a ghost target. Indeed, if we suppose that the depth of the target is $z_{As}$ (whereas the true value is $z_{T}$), then we have $\beta =\frac{2D-\left(z_{As}+z_{O}\right)}{2D-\left(z_{T}+z_{O}\right)}$, whose biggest value $\beta_{Max}=\frac{2D-z_{O}}{2D-\left(z_{T}+z_{O}\right)}$, and the minimum value is $\beta_{Min}=\frac{2D-\left(z_{Max}+z_{O}\right)}{2D-\left(z_{T}+z_{O}\right)}$, where $z_{Max}$ is the largest depth of a submarine vehicle. Typically, in deep water, D≥4000 m. A reasonable choice of $z_{Max}$ could be 400 m. We can then have a range of β: $\left[\beta_{Min},\beta_{Max}\right]=\left[\frac{7600-z_{O}}{8000-\left(z_{T}+z_{O}\right)},\frac{8000-z_{O}}{8000-\left(z_{T}+z_{O}\right)}\right]$. For instance, when the depths of the antenna and the target are, respectively, 200 and 100 m, we have $\left[\beta_{Min},\beta_{Max}\right]=\left[0.974,1.013\right]$. In this way, we bound the set of ghost targets, and we can expect that the bias induced by a wrong choice of $z_{As}$ is very low.*
*4.* 
*For case (4) again, with a direct path, if the target is not at the same depth as the antenna, $\beta =\frac{z_{As}-z_{O}}{z_{T}-z_{O}}$. Because β is a positive number, $z_{As}-z_{O}$ has the same sign as $z_{T}-z_{O}$: if $z_{T}>z_{O}$, then $z_{O}<z_{As}\leq z_{Max}$, and $\left[\beta_{Min},\beta_{Max}\right]=\left[0,\frac{z_{Max}-z_{O}}{z_{T}-z_{O}}\right]$; if $z_{T}<z_{O}$, then $0\leq z_{As}<z_{O}$, and $\left[\beta_{Min},\beta_{Max}\right]=\left[0,\frac{z_{O}}{z_{T}-z_{O}}\right]$. In both cases, the range $\left[\beta_{Min},\beta_{Max}\right]$ is too wide to be useful.*

*If the target and the antenna are at the same depth,*
β
*can take any positive value.*



### 3.2. Estimation of the Trajectory

We run 500 Monte Carlo simulations for a typical scenario described as follows:

The observer starts from ${\left(\begin{array}{cc} 0 & 0 \end{array}\right)}^{T}$ at the depth $z_{O}=200\;$m. Its speed and heading are, respectively, 5 m/s and 0°. The initial position of the target is ${\left(\begin{array}{cc} 5000 & 7000 \end{array}\right)}^{T}$ and its depth is $z_{T}=100\;$m. Its route is 45° and its speed is 4 m/s.
The measurements are collected every 4 s (Δt=4 s). The scenario lasts 20 min.The sea bottom depth is 4000 m. The detected ray is a bottom-reflected ray.The assumed target depth is $z_{As}=200\;$m (whereas the true one is 100 m).First, the measurements have been corrupted with an additive Gaussian noise whose standard deviation is $\sigma =1.7\times 10^{-2}$.

Then, we choose the least squares estimator, which is identical to the maximum likelihood estimator with these assumptions. Note that, in open literature about TMA, the confidence regions are given by the confidence ellipsoid obtained with the covariance matrix of the estimate. Since the maximum likelihood estimate is asymptotically efficient under nonrestrictive conditions, we use here the Cramér–Rao lower bound to compute such confidence regions.

The result of the simulation is presented in Table 1 and illustrated in Figure 4. Obviously, even if the assumption made on the target’s depth makes the state vector observable, it remains inestimable: the hugeness of the diagonal elements of the CRLB does not allow this kind of TMA to be employed. We note in Figure 4 that the cloud of horizontal estimates is hyperbola-shaped. This is because the state vector is “weakly” estimable. The parametric equation of this hyperbola is
$$\{\begin{array}{c} x\left(\omega \right)=\zeta_{As}\sinh\left(\omega \right)\;\\ y\left(\omega \right)=\frac{\zeta_{As}m}{\sqrt{1-m^{2}}}\cosh\left(\omega \right) \end{array},\;\mathrm{with}\;\zeta_{As}=2D-\left(z_{As}+z_{O}\right),\;\mathrm{and}\;m=\frac{y_{OT}\left(0\right)}{\sqrt{x_{OT}^{2}\left(0\right)+y_{OT}^{2}\left(0\right)+\zeta_{OT}^{2}}}$$

We further reduced the standard deviation to $\sigma =1.7\times 10^{-4}$ in order to appreciate the behavior of the MLE. With this (unrealistic) value, the MLE is efficient, as shown in Table 2 and in Figure 5 (which validates our observability analysis).

Our conclusion is that the state vector is not estimable, even though it is observable with an assumption on the target’s depth.

This is why we propose to maneuver the antenna in order to render the state vector observable with no assumption on the target’s depth, and to augment the information about it.

## 4. TMA with One Ray When the Array Maneuvers

In this section, the antenna maneuvers, i.e., it changes its own heading. We start by proving that the state vector is observable (without any assumption on the target’s depth). Then, we have recourse to perform Monte Carlo simulations to evaluate the performance of the MLE.

### 4.1. Observability Analysis

**Theorem** **2.**
*Suppose the antenna’s trajectory is composed of two successive legs at constant velocity (however with the same speed). Let the target be in CV motion. The linear array acquires the conical angles of the wave emitted from the target, the path of the ray being known as well as the sea bottom depth. If the target is broadside or endfire to the antenna during a leg, then there is at most a ghost target. Otherwise, there is no ghost target.*


Due to its length, the proof of this theorem is given in the Appendix A.

### 4.2. Estimation

In this subsection, we present the result of 500 Monte Carlo simulations that are run to illustrate the behavior of the proposed estimators. First, we give the scenario used here.

The center of the array and the initial position of the source are, respectively, at ${\left(\begin{array}{ccc} 0 & 0 & 200 \end{array}\right)}^{T}$ and ${\left(\begin{array}{ccc} 5000 & 7000 & 100 \end{array}\right)}^{T}$at the very beginning of the scenario. The speed of the array is a constant along the scenario and is equal to 5 m/s. The trajectory of the array is composed of two legs linked by an arc of a circle. The first leg lasts 1 min 40 s, during which the array’s heading is 135°. Then, the array turns to the right with a turn rate equal to 20°/min to adopt a new heading equal to 270°. The duration of the maneuver is hence equal to 6 min 44 s. The second leg lasts 5 min, so the total duration of the scenario is 13 min and 20 s. Meanwhile, the target is navigating with a heading equal to 45° and a speed of 4 m/s. The bottom depth is D=4000 m.

The state vector we have to estimate is hence $X={\left(\begin{array}{ccccc} 5000 & 7000 & 100 & 2.83 & 2.83 \end{array}\right)}^{T}$.

The array is assumed to measure the cosines of the conical angles of the bottom-reflected path given by
$$m\left(t_{k}\right)=\frac{y_{OT}\left(t_{k}\right)}{\sqrt{x_{OT}^{2}\left(t_{k}\right)+y_{OT}^{2}\left(t_{k}\right)+{\left[2D-\left(z_{T}+z_{O}\right)\right]}^{2}}}+\varepsilon_{k}$$

Measurements are acquired every Δt=4 s, with $t_{k}=\left(k-1\right)\Delta t$.

The noise vector $\varepsilon_{k}$ is assumed to be Gaussian, 0-mean and its standard deviation equal to $\sigma =1.7\times 10^{-2}$. The vectors $\varepsilon_{k}$ are also assumed to be temporally independent.

Again, we choose the least squares estimator.

#### 4.2.1. Estimation of X

The 500 obtained estimates of the initial horizontal position are plotted in Figure 6, together with the trajectory of the target, the 90%-confidence ellipse and the trajectory of the array. Again, the view is from the sky.

The performance of the estimator (bias and standard deviation of each component) is presented in Table 3.

A convenient way to evaluate the behavior of an estimator is to compute the so-called normalized estimation error squared (NEES) [27], defined as $N_{l}={\left({\hat{X}}_{l}-X\right)}^{T}F\left({\hat{X}}_{l}-X\right)$, where F is the FIM, and ${\hat{X}}_{l}$ is the estimate computed at the l-th simulation. If ${\hat{X}}_{l}$ is Gaussian-distributed with X as the mathematical expectation and the CRLB as the covariance matrix, then $N_{l}$ is chi-square distributed with d degrees of freedom ($\chi_{d}^{2}$), where *d* is the dimension of X (here 5). From the central limit theorem, the averaged NEES $N_{S}≜\frac{1}{N_{Sim}}\sum_{l=1}^{N_{Sim}}N_{l}$ is approximately Gaussian; its mathematical expectation is *d*, and its standard deviation is equal to $\sqrt{\frac{2d}{N_{Sim}}}$.

From our simulations, we obtain $N_{S}=5.34$.

In conclusion, the estimator can be declared efficient. However, the minimum standard deviation of the target’s depth is not compatible with the physical constraints: with the standard deviation given in Table 3, the target could be up above the sea surface! Therefore, a palliative of this is to impose a depth on the target. Indeed, we saw in Section 3.1 that a supposed depth creates a small bias in estimation of the horizontal position of the target.

#### 4.2.2. Estimation of X Reduced When the Depth of the Target Is Fixed

Now, the third component of X does not have to be estimated. The new state vector is the denoted as $X_{r}={\left(\begin{array}{cccc} x_{T}\left(0\right) & y_{T}\left(0\right) & {\dot{x}}_{T} & {\dot{y}}_{T} \end{array}\right)}^{T}$. We impose that $z_{As}=200\;$m (whereas the true depth is still 100 m). Hence, we introduce a bias.

Figure 7 displays the position’s estimates in the same manner as Figure 6. The bias is not visible to the naked eye. However, Table 4 reveals this bias, which may be acceptable in a real situation. Even though the averaged NEES (=7.31) is out of its 90% confidence interval, its value remains acceptable.

The main interest of assuming the depth to be known is to economize on the CPU time, and reduce the standard deviation of the remaining components to estimate. We are in the presence of the well-known bias–variance tradeoff.

#### 4.2.3. Estimation of the Reduced State Vector by the Conventional BOTMA

In such a scenario, the conventional BOTMA can be run by neglecting the site effect, so by imposing that cos(ϕ(t))=1, ∀t. The (incorrect) noise-free measurement model is then
cos(α(t))=cos(θ(t)−h(t)).

The results are plotted in Figure 8. Obviously, a huge bias appears, leading to an averaged NEES equal to 1960. More precisely, the bias on the components of the reduced state vector is ${\left(\begin{array}{cccc} -3062.8 & -2319.9 & 14.4 & 15.8 \end{array}\right)}^{T}$, rendering the BOTMA inoperative. Clearly, the conventional BOTMA cannot be recommended for the near field. This justifies a posteriori the interest in taking the site effect and the nature of the wave ray into account, as previously pointed out in the introduction of [23].

## 5. TMA from the Direct Path and the Bottom-Reflected Path

We assume in this section that the sound wave emitted by the target travels on the direct path and the bottom-reflected path.

### 5.1. Observability

**Theorem** **3.**
*Let a linear antenna and a source both be in CV motion.*

*The antenna acquires the cosines of the conical angles of the direct path and of the bottom-reflected path.*
*1.* 
*If the target is broadside to the array, then the set of ghost targets is uncountable: it is composed of all the (virtual) targets at broadside to the array.*
*2.* 
*If the target is endfire to the antenna, the set of ghost targets is composed of virtual sources at endfire to the antenna.*
*3.* 
*If the route of the antenna and the route of the target are parallel, then the set of ghost targets is uncountable: at each depth $z_{G}$, there are two ghost targets moving on a cylinder whose axis is the antenna axis, and the radius is a positive scalar $\beta =\sqrt{\frac{D-z_{G}}{D-z_{T}}}$. The relative ghost target velocity is equal to β times the target’s velocity. The initial distance between the ghost image and the center of the antenna is equal to β times the initial distance between the ghost image and the center of the antenna.*
*4.* 
*If the route of the antenna and the route of the target are not parallel, then there are three ghost targets whose motion relative to the antenna is $P_{OG}\left(t\right)=SP_{OT}\left(t\right)$,$\;P_{OG}\left(t\right)=\beta P_{OT}\left(t\right)$, and $P_{OG}\left(t\right)=\beta SP_{OT}\left(t\right)$, where S is the matrix of the axial symmetry around the line of the antenna, and $\beta ≜\frac{D-z_{O}}{D-z_{T}}$. If the depth of the antenna is equal to the depth of the source, then there is one single ghost target given by $P_{OG}\left(t\right)=SP_{OT}\left(t\right)$.*



**Proof of Theorem 3.** With no loss of generality, we will again assume that the axis of the sensor array is pointed toward north and that the target is in the half-space where the second component y of any vector is positive. A convenient rotation helps us in this case. So the noise-free measurements at time *t* are $m_{1}\left(t\right)=\frac{y_{OT}\left(t\right)}{\sqrt{x_{OT}^{2}\left(t\right)+y_{OT}^{2}\left(t\right)+z_{OT}^{2}}}$, and $m_{2}\left(t\right)=\frac{y_{OT}\left(t\right)}{\sqrt{x_{OT}^{2}\left(t\right)+y_{OT}^{2}\left(t\right)+{\left[2D-\left(z_{T}+z_{O}\right)\right]}^{2}}}$.We have to seek a five-dimensional state vector $X_{G}={\left(\begin{array}{ccccc} x_{G}\left(0\right) & y_{G}\left(0\right) & z_{G} & {\dot{x}}_{G} & {\dot{y}}_{G} \end{array}\right)}^{T}$ defining the trajectory of a ghost target, i.e., producing the same noise-free measurement as X, that is $m_{1}\left(t\right)=\frac{y_{OG}\left(t\right)}{\sqrt{x_{OG}^{2}\left(t\right)+y_{OG}^{2}\left(t\right)+z_{OG}^{2}}}$, and $m_{2}\left(t\right)=\frac{y_{OG}\left(t\right)}{\sqrt{x_{OG}^{2}\left(t\right)+y_{OG}^{2}\left(t\right)+{\left[2D-\left(z_{G}+z_{O}\right)\right]}^{2}}}$.hence satisfying the two following equalities (in time):(6)$$\frac{y_{OT}\left(t\right)}{\sqrt{x_{OT}^{2}\left(t\right)+y_{OT}^{2}\left(t\right)+z_{OT}^{2}}}=\frac{y_{OG}\left(t\right)}{\sqrt{x_{OG}^{2}\left(t\right)+y_{OG}^{2}\left(t\right)+z_{OG}^{2}}}$$
(7)$$\frac{y_{OT}\left(t\right)}{\sqrt{x_{OT}^{2}\left(t\right)+y_{OT}^{2}\left(t\right)+{\left[2D-\left(z_{T}+z_{O}\right)\right]}^{2}}}=\frac{y_{OG}\left(t\right)}{\sqrt{x_{OG}^{2}\left(t\right)+y_{OG}^{2}\left(t\right)+{\left[2D-\left(z_{G}+z_{O}\right)\right]}^{2}}}$$
under the constraint that $z_{G}$ is in [0,D].Case 1: $y_{OT}\left(t\right)$ is a zero function, i.e., $\forall t\;y_{OT}\left(t\right)=0$.The target is broadside to the antenna, so any ghost targets will be too (see Case 1 in the proof of theorem 1).Case 2: $y_{OT}\left(t\right)$ is not a zero function.From Case 2 of the proof of theorem 1, there is a positive scalar β such that $y_{OG}\left(t\right)=\beta y_{OT}\left(t\right)$.
(8)$$\left(6\right)\Leftrightarrow \sqrt{x_{OG}^{2}\left(t\right)+y_{OG}^{2}\left(t\right)+z_{OG}^{2}}=\beta \sqrt{x_{OT}^{2}\left(t\right)+y_{OT}^{2}\left(t\right)+z_{OT}^{2}}\;\Leftrightarrow \left[x_{OG}^{2}\left(t\right)+y_{OG}^{2}\left(t\right)+z_{OG}^{2}\right]=\beta^{2}\left[x_{OT}^{2}\left(t\right)+y_{OT}^{2}\left(t\right)+z_{OT}^{2}\right]$$
(9)$$\left(7\right)\Leftrightarrow \left[x_{OG}^{2}\left(t\right)+y_{OG}^{2}\left(t\right)+{\left[2D-\left(z_{G}+z_{O}\right)\right]}^{2}\right]\;=\beta^{2}\left[x_{OT}^{2}\left(t\right)+y_{OT}^{2}\left(t\right)+{\left[2D-\left(z_{T}+z_{O}\right)\right]}^{2}\right]$$
Subtracting (9) from (8), we get $z_{OG}^{2}-{\left[2D-\left(z_{G}+z_{O}\right)\right]}^{2}=\beta^{2}\left[z_{OT}^{2}-{\left[2D-\left(z_{T}+z_{O}\right)\right]}^{2}\right]$.Now, we simplify the expressions of these two terms:$$z_{OG}^{2}-{\left[2D-\left(z_{G}+z_{O}\right)\right]}^{2}=-4\left(D-z_{G}\right)\left(D-z_{O}\right)$$
$$z_{OT}^{2}-{\left[2D-\left(z_{T}+z_{O}\right)\right]}^{2}=-4\left(D-z_{T}\right)\left(D-z_{O}\right)$$
We deduce from this that
(10)$$\beta =\sqrt{\frac{D-z_{G}}{D-z_{T}}}$$
Note that β=1 iif $z_{G}=z_{T}$.
(11)$$\left(8\right)\Leftrightarrow x_{OG}^{2}\left(t\right)+y_{OG}^{2}\left(t\right)+z_{OG}^{2}=\beta^{2}\left[x_{OT}^{2}\left(t\right)+y_{OT}^{2}\left(t\right)+z_{OT}^{2}\right]\;\Leftrightarrow x_{OG}^{2}\left(t\right)-\beta^{2}x_{OT}^{2}\left(t\right)=\beta^{2}z_{OT}^{2}-z_{OG}^{2}$$
Since $x_{OG}^{2}\left(t\right)-\beta^{2}x_{OT}^{2}\left(t\right)$ is a polynomial function of degree 2, (11) is equivalent to
(12)$$x_{OG}^{2}\left(0\right)=\beta^{2}x_{OT}^{2}\left(0\right)+\beta^{2}z_{OT}^{2}-z_{OG}^{2}$$
(13)$$x_{OG}\left(0\right){\dot{x}}_{OG}=\beta^{2}x_{OT}\left(0\right){\dot{x}}_{OT}$$
(14)$${\dot{x}}_{OG}^{2}=\beta^{2}{\dot{x}}_{OT}^{2}$$
First case ${\dot{x}}_{OT}=0$Equation (14) implies that ${\dot{x}}_{OG}=0$.Consequently, for any $z_{G}$ in [0,D], the vector $X_{OG}={\left(\begin{array}{ccccc} \pm \sqrt{\beta^{2}x_{OT}^{2}\left(0\right)+\beta^{2}z_{OT}^{2}-z_{OG}^{2}} & \beta y_{OT}\left(0\right) & z_{OG} & 0 & \beta {\dot{y}}_{OT} \end{array}\right)}^{T}$ (with $\beta =\sqrt{\frac{D-z_{G}}{D-z_{T}}}$) defines the trajectory of a ghost target.Second case ${\dot{x}}_{OT}\neq 0$Using (14), and squaring the terms of (13), we get $x_{OG}^{2}\left(0\right)=\beta^{2}x_{OT}^{2}\left(0\right)$.Reporting this in (12), we obtain finally $\beta^{2}z_{OT}^{2}=z_{OG}^{2}$, i.e.,
(15)$$\beta^{2}={\left(\frac{z_{OG}}{z_{OT}}\right)}^{2}$$
If $z_{T}=z_{O}$, then $z_{G}=z_{O}$. In this case, β=1, and consequently $y_{OG}\left(t\right)=y_{OT}\left(t\right)$ and $x_{OG}^{2}\left(t\right)=x_{OT}^{2}\left(t\right)$ from (11). The source’s trajectory is observable up to the axial symmetry around the (Oy)-axis.Equations (10) and (15) give us $\frac{D-z_{G}}{D-z_{T}}={\left(\frac{z_{OG}}{z_{OT}}\right)}^{2}$.The unknown $z_{G}$ is hence a root of the following equation of degree 2:${\left(z_{G}-z_{O}\right)}^{2}-\frac{{\left(z_{T}-z_{O}\right)}^{2}}{D-z_{T}}\left(D-z_{G}\right)=0$ which can be expanded as follows: $z_{G}^{2}+z_{G}\left[-2z_{O}+\frac{{\left(z_{T}-z_{O}\right)}^{2}}{D-z_{T}}\right]-\frac{D{\left(z_{T}-z_{O}\right)}^{2}}{D-z_{T}}+z_{O}^{2}=0$.Of course, $z_{T}$ is a root of this equation. For this value, $z_{G}=z_{T}$, hence β=1.The second root ($z_{G}$ itself) is hence $2z_{O}-z_{T}-\frac{{\left(z_{T}-z_{O}\right)}^{2}}{D-z_{T}}≜z_{G}$. We can check readily that $z_{G}-z_{O}=z_{O}-z_{T}-\frac{{\left(z_{T}-z_{O}\right)}^{2}}{D-z_{T}}=\frac{\left(z_{O}-z_{T}\right)\left(D-z_{T}\right)-{\left(z_{T}-z_{O}\right)}^{2}}{D-z_{T}}$.Hence, $\frac{z_{G}-z_{O}}{z_{T}-z_{O}}=\frac{z_{O}-D}{D-z_{T}}$ (which is negative).We deduce from this that:
when the target’s depth is larger than the array’s depth, there is a ghost whose depth is smaller than the array’s depth, and vice versa.β, which is a positive coefficient, is equal to $\frac{D-z_{O}}{D-z_{T}}$, or 1.Therefore, we have identified three ghost targets:the first one is defined by $X_{OG}={\left(\begin{array}{ccccc} -x_{OT}\left(0\right) & y_{OT}\left(0\right) & z_{OT} & -{\dot{x}}_{OT} & {\dot{y}}_{OT} \end{array}\right)}^{T}$,the second is defined by $X_{OG}={\left(\begin{array}{ccccc} \beta x_{OT}\left(0\right) & \beta y_{OT}\left(0\right) & -\beta z_{OT} & \beta {\dot{x}}_{OT} & {\dot{\beta y}}_{OT} \end{array}\right)}^{T}$,and the third by $X_{OG}={\left(\begin{array}{ccccc} -\beta x_{OT}\left(0\right) & \beta y_{OT}\left(0\right) & -\beta z_{OT} & -\beta {\dot{x}}_{OT} & {\dot{\beta y}}_{OT} \end{array}\right)}^{T}$. □

**Remark** **2.**
*Most of the time, the depth of a submarine vehicle is under the operational constraint: values of*
$z_{T}$
*are in*
$\left[0,z_{Max}\right]$
*and*
$z_{Max}\ll D$
*. For example,*
$z_{Max}=400\;$
*m, while*
D=4000
*m.*

*The proof of the previous theorem must be adapted to this new constraint.*
*First, we use the fact that the function*$u\mapsto f\left(u\right)≜2z_{O}-u-\frac{{\left(u-z_{O}\right)}^{2}}{D-u}$*is an involution, i.e.,*f(f(u))=u.*Since* $f\left(0\right)=2z_{O}-\frac{z_{0}^{2}}{D}$*,* $f\left(2z_{O}-\frac{z_{0}^{2}}{D}\right)=0$.
*Now the question is: what are the values of*
$z_{O}$
*for which the following inequality holds:*
$2z_{O}-\frac{z_{0}^{2}}{D}\leq z_{Max}$
*, the greatest value of*
$z_{O}$
*guaranteeing that*
$2z_{O}-\frac{z_{0}^{2}}{D-z_{T}}\leq z_{Max}$
*is*
$D-D\sqrt{1-\frac{z_{Max}}{D}}$
*(which is less than*
$z_{Max}$
*).*
*If*$z_{O}>D-D\sqrt{1-\frac{z_{Max}}{D}}$*, then*$z_{G}>z_{Max}$*. In this case, there is a unique ghost target given by*$X_{G}={\left(\begin{array}{ccccc} -x_{T}\left(0\right) & y_{T}\left(0\right) & z_{T} & -{\dot{x}}_{T} & {\dot{y}}_{T} \end{array}\right)}^{T}$.
*If*
$z_{O}\leq D-D\sqrt{1-\frac{z_{Max}}{D}}$
*, then*
$z_{G}\leq z_{Max}$
*. In this case, there are three ghost targets:*

*one is defined by*
$X_{G}={\left(\begin{array}{ccccc} -x_{T}\left(0\right) & y_{T}\left(0\right) & z_{T} & -{\dot{x}}_{T} & {\dot{y}}_{T} \end{array}\right)}^{T}$
*,*

*the second is defined by*
$X_{G}={\left(\begin{array}{ccccc} \beta x_{T}\left(0\right) & \beta y_{T}\left(0\right) & f\left(z_{T}\right) & \beta {\dot{x}}_{T} & \beta {\dot{y}}_{T} \end{array}\right)}^{T}$
*,*
*and the third by*$X_{G}={\left(\begin{array}{ccccc} -\beta x_{T}\left(0\right) & \beta y_{T}\left(0\right) & f\left(z_{T}\right) & -\beta {\dot{x}}_{T} & \beta {\dot{y}}_{T} \end{array}\right)}^{T}$.
*Note that the operational constraint allows us to benefit from the following range:*

$\frac{D-z_{Max}}{D}\leq \beta \leq \frac{D}{D-z_{Max}}$
*. For example, when*
$z_{Max}=\frac{D}{10}$
*,*
0.9≤β≤1.11
*. Consequently, the ghost target is very close to the target of interest.*


### 5.2. Estimation of the Trajectory

This section is devoted to the estimation of the target’s trajectory, or in other words, the estimation of X with $t^{*}=0$ (the first time). Before going into detail, we compute the so-called Cramér–Rao lower bound to evaluate the asymptotical performance of any unbiased estimator.

We have considered two typical scenarios. In both, the array is assumed motionless (or, more realistically, all the mobiles are referenced to it) at the depth $z_{O}=200\;$m, and the state vector defining the target’s trajectory is given by the state vector $X={\left(\begin{array}{ccccc} 5000 & 7000 & 100 & 2.83 & 2.83 \end{array}\right)}^{T}$. The standard deviation of the measurement is $\sigma =1.7\times 10^{-2}$. The total duration of the scenario is 5 min, and the sampling time is Δt=4 s; consequently, the number of measurement couples is N=75.

In the first scenario, the bottom depth is D=2000 m, while in the second, D=4000 m.

Note that in the first scenario, β=0.89, and in the second one, β=0.97. The ghost target is hence very close to the target of interest.

#### 5.2.1. Estimability

As pointed out in Section 1, the state vector X is “estimable” if its asymptotical performance given by the CRLB is compatible with the physical constraints. Typically, if the minimum standard deviation defined by the square root of the third diagonal element of the CRLB (hence of the depth) is much larger than the depth, then X is declared non-estimable.

First scenario

For this scenario, the square root of the diagonal of the CRLB $\sigma_{CRLB}={\left(\begin{array}{ccccc} 1.16\;\times 10^{6} & 1.59\;\times 10^{6} & 8.22\;\times 10^{5} & 637.9 & 646.1 \end{array}\right)}^{T}$.

2.Second scenario

With the bottom depth, things are not much better, since $\sigma_{CRLB}={\left(\begin{array}{ccccc} 6.59\;\times 10^{5} & 8.96\;\times 10^{5} & 9.73\;\times 10^{5} & 352.6 & 362.2 \end{array}\right)}^{T}$.

In both cases, the minimum standard deviations are huge. We can conclude that the state vector is not estimable. Computations of minimum standard deviations were made for various scenarios; in all, the state vector is not estimable.

A palliative of this is to fix the depth of the source at an arbitrary and realistic value, say $z_{As},$ and compute the CRLB of the reduced state vector $X_{r}≜{\left(\begin{array}{cccc} x_{T}\left(0\right) & y_{T}\left(0\right) & {\dot{x}}_{T} & {\dot{y}}_{T} \end{array}\right)}^{T}$ when we assume that $z_{T}=z_{As}$. For example, for $z_{As}=300$ m, the minimum standard deviations are
$$\sigma_{CRLB}={\left(\begin{array}{cccc} 281.17 & 319.37 & 1.78 & 2.02 \end{array}\right)}^{T}\;\mathrm{for}\;\mathrm{the}\;\mathrm{first}\;\mathrm{scenario},\;\mathrm{and}$$
$$\sigma_{CRLB}={\left(\begin{array}{cccc} 130.1 & 115.3 & 0.80 & 0.71 \end{array}\right)}^{T}\;\mathrm{for}\;\mathrm{the}\;\sec\mathrm{ond}\;\mathrm{one}.$$

Therefore, we propose to estimate the state vector with this hypothesis ($z_{As}=300$ m). In so doing, we introduce a bias. The next subsection gives us the result of the 500 Monte Carlo simulations.

#### 5.2.2. Monte Carlo simulations

The computation of the maximum likelihood estimator (MLE) is made with the Gauss‒Newton routine. No numerical issue was encountered.

First scenario

The performance of the MLE is summarized in Table 5. We have numerically computed the bias and the empirical standard deviation (given, respectively in the second and third column of the table). We can see that the empirical standard deviation is very close to that given by the CRLB. However, as expected, the MLE is biased (of course, there is no bias if we choose $z_{As}=z_{T}$). In Figure 9, the 90% confidence ellipse is drawn, together with the cloud of the 500 estimates (in pink).

2.Second scenario: Bottom depth D=4000 m.

Again, the performance is presented in Table 6. The bias of the estimator is similar to the one obtained for the first scenario. Only the empirical standard deviations of ${\left(\begin{array}{cc} x_{T}\left(0\right) & y_{T}\left(0\right) \end{array}\right)}^{T}$ are larger than that computed from the CRLB. However, Figure 10 shows us that the cloud of estimates is close to the true value and not spread.

What is remarkable is the short duration and still the very good performance (in terms of accuracy) of the result. Numerous simulations (not reported here) were performed; all confirm the correct performance of the MLE. The shortness of the scenario is crucial, because everything that we propose here works properly under the condition that the sea bottom is a plane. During a short scenario, this assumption is likely.

## 6. Conclusions

In this paper, conical-angle TMA has been addressed, and various multipaths of sound have been taken into account. The sensor is a line array. Observability was analyzed deeply, allowing all the existing ghost targets to be identified. The main results are that, if the array detects one ray (corresponding to one path), the trajectory is not observable: the set of ghost targets is composed of trajectories that are homothetic to the trajectory of the target of interest, and their symmetrical images by the axial symmetry around the line array. If the array detects two rays (corresponding to two different paths), the number of ghost targets is reduced to three (except when the target is endfire or broadside to the antenna). When the antenna maneuvers, the target’s trajectory is observable (apart from the special scenario where there is one single ghost target). Even for “observable” scenarios, the depth of the target is not estimable (its asymptotical standard deviation is huge). In these cases, we give a non-restrictive palliative that allows us to provide estimates close to the truth.

In the future, in this context, many problems remain to be faced: identification of the paths, maneuvering targets, and fusion of data collected by other sensors, as in [28]. The problem of seeking a “good” maneuver of the observer, as it was solved in a 2D environment [29,30,31], will be addressed in the future. Some of these problems are already under investigation.

## Figures and Tables

**Figure 1 sensors-21-04797-f001:**
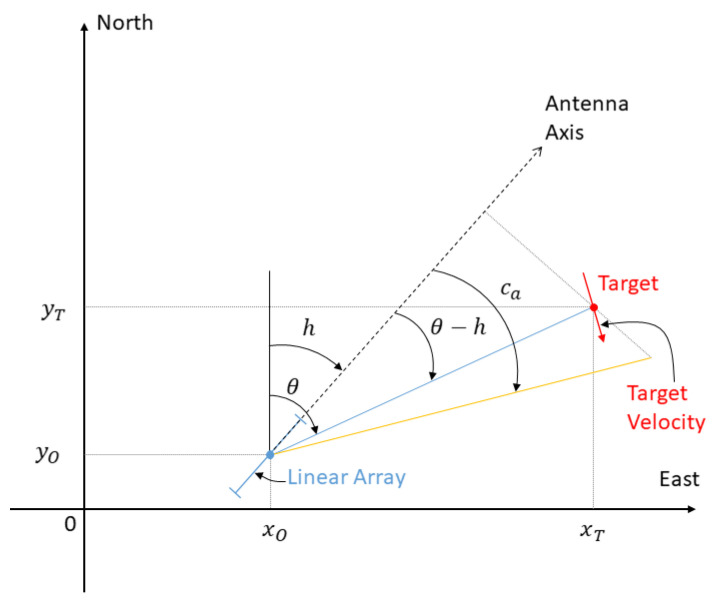
A typical scenario, viewed from the sky.

**Figure 2 sensors-21-04797-f002:**
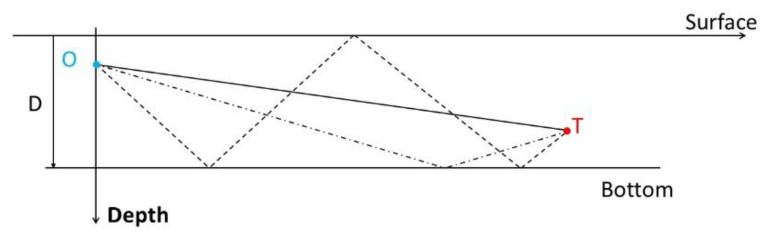
Three examples of ray paths: the solid line represents the direct path $\left(\delta ,n_{B},n_{S}\right)=\left(+1,0,0\right)$, the dashed-dotted line represents the bottom reflected path $\left(\delta ,n_{B},n_{S}\right)=\left(+1,1,0\right)$, and the dashed line represents the bottom-surface-bottom reflected path $\left(\delta ,n_{B},n_{S}\right)=\left(+1,2,1\right)$.

**Figure 3 sensors-21-04797-f003:**
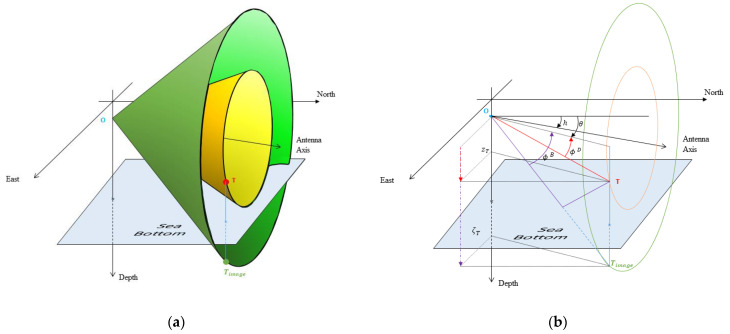
Cones of ambiguity. (**a**) The cones that the target belongs to, and the one that the image-target belongs to. (**b**) Example of conical angles of the target and of the image-target, for a bottom-reflected ray: $\phi^{D}$ and $\phi^{B}$ are the elevations of the direct path and of the bottom-reflected path, respectively.

**Figure 4 sensors-21-04797-f004:**
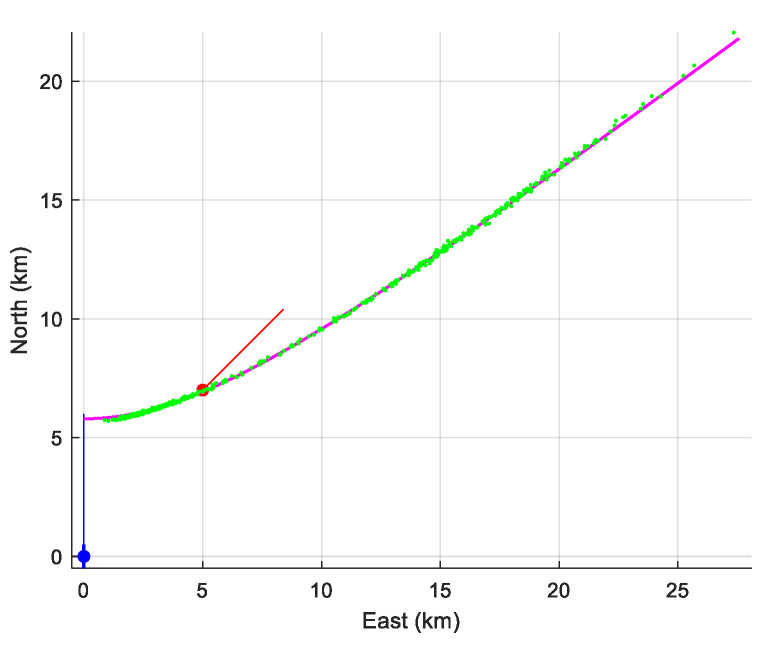
The cloud of estimated position (in green), a piece of the hyperbola (intersection of the cone of ambiguity and the plane $z=z_{As}\left(=200\;m\right)$, for $\sigma =1.7\;\times 10^{-2}$.

**Figure 5 sensors-21-04797-f005:**
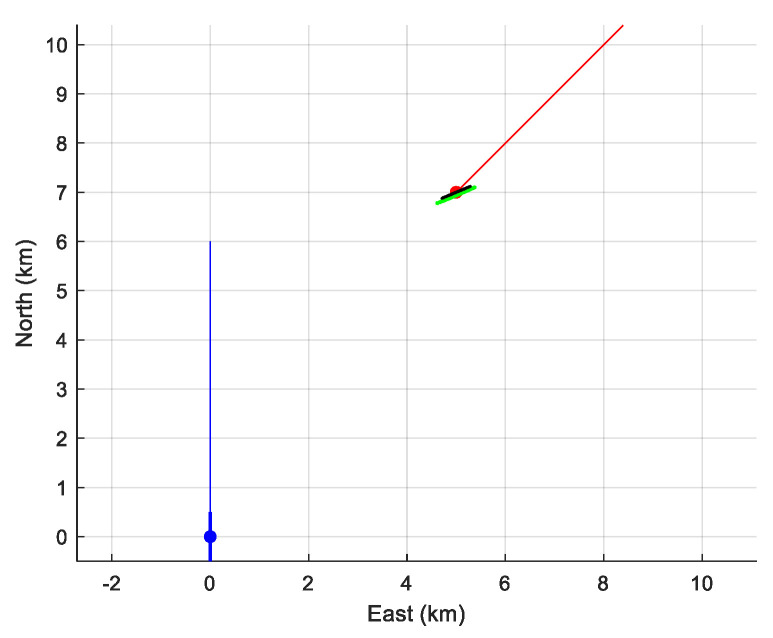
The cloud of estimated position (in green) for $\sigma =1.7\times 10^{-4}$. The cloud is no longer hyperbola-shaped. The small black segment is the 90%-confidence ellipsoid.

**Figure 6 sensors-21-04797-f006:**
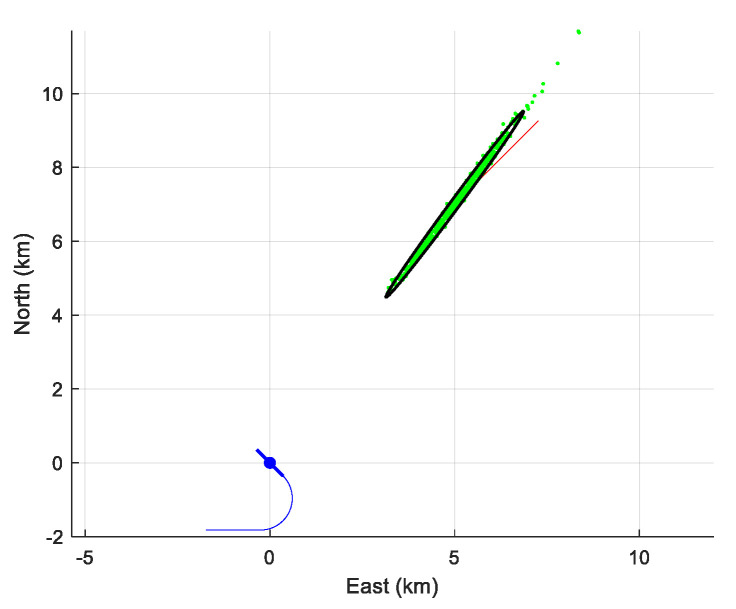
The cloud of the 500 initial positions estimates and the 90%-confidence ellipse.

**Figure 7 sensors-21-04797-f007:**
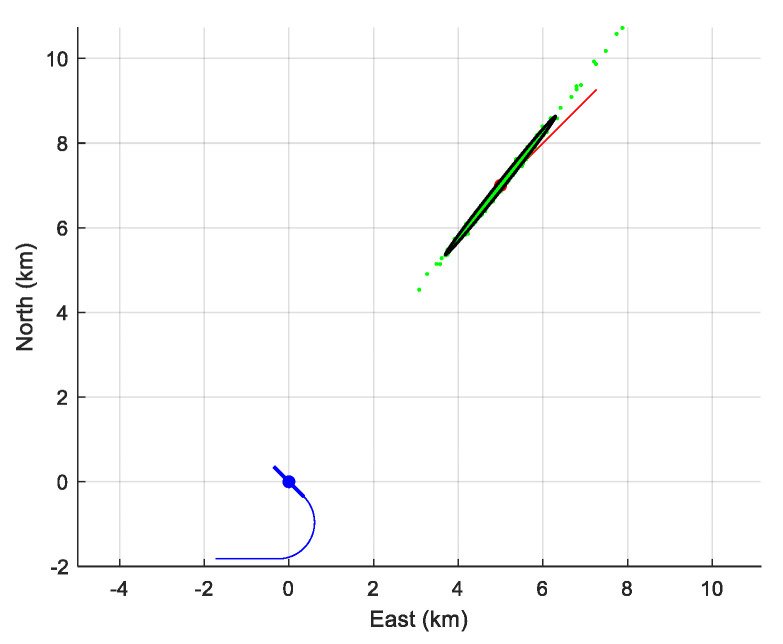
The cloud of the 500 initial positions estimates with the reduced state vector and the 90%-confidence ellipse.

**Figure 8 sensors-21-04797-f008:**
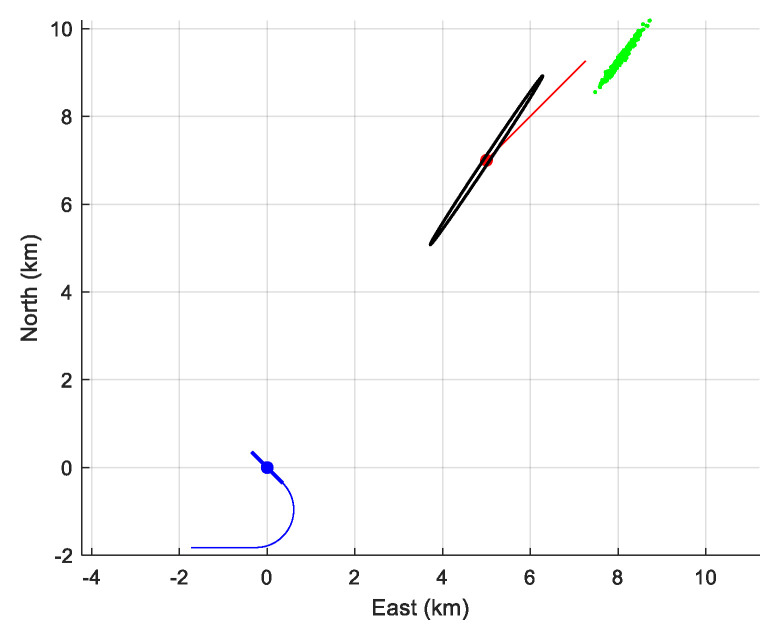
The cloud (in green) of the 500 initial positions estimates given by the classic BOTMA together with the 90%-confidence ellipse.

**Figure 9 sensors-21-04797-f009:**
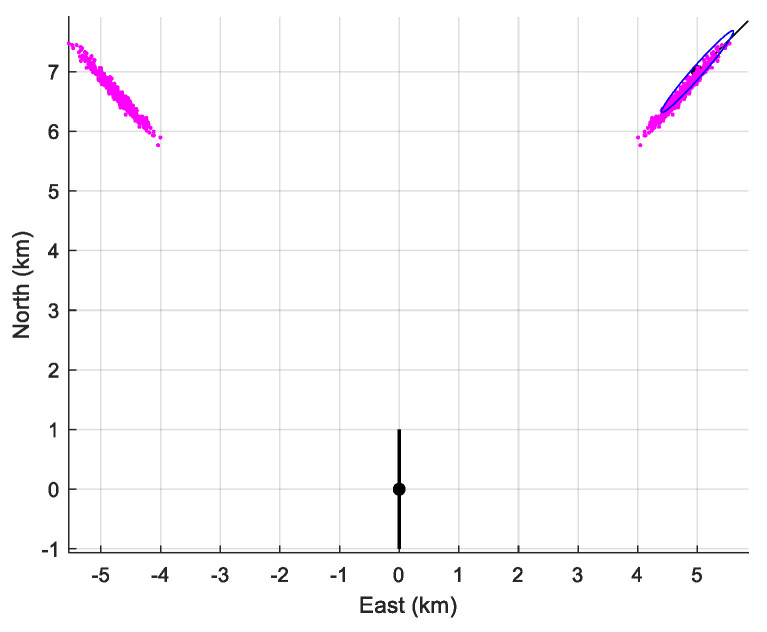
The location of the sensor array (in black), the cloud of the 500 estimates and the 90%-confidence ellipse when D=2000 m, $z_{As}=300\;$m, and $z_{T}=100\;$m. The symmetrical cloud is plotted too.

**Figure 10 sensors-21-04797-f010:**
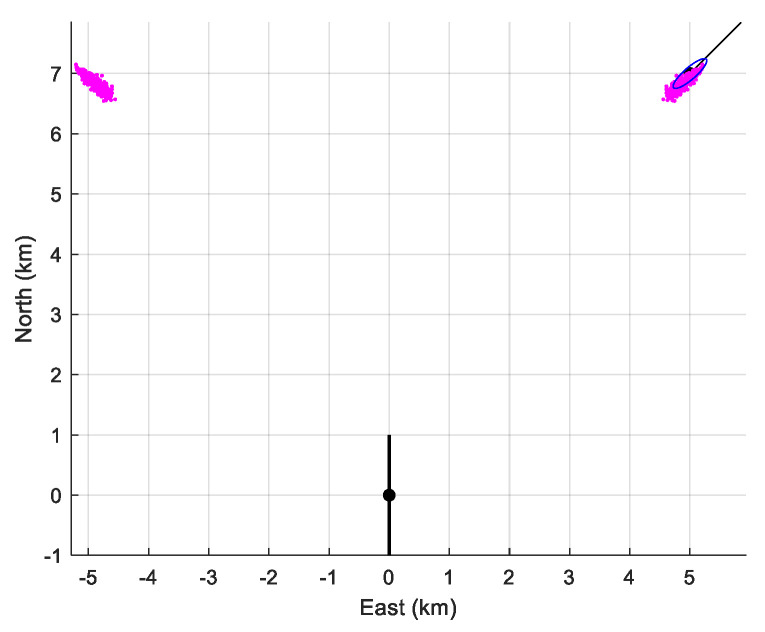
The location of the sensor array (in black), the cloud of the 500 estimates of the initial positions, and the 90%-confidence ellipse when D=4000 m, $z_{As}=300\;$ m, and $z_{As}=100\;$ m, together with the symmetrical cloud.

**Table 1 sensors-21-04797-t001:** Performance of the estimator of the reduced state vector when $\sigma =1.7\times 10^{-2}$, in terms of bias, sample standard deviation and the one given by the square root of the diagonal of the CRLB.

$X_{r}$	Bias	$\sigma_{samp}$	$\sigma_{CRLB}$
5000 m	−3525	6962	13,356
7000 m	−2367	4052	5599
2.83 m/s	−1.37	1.81	4.35
2.83 m/s	0.53	1.62	2.75

**Table 2 sensors-21-04797-t002:** Performance of the estimator of the reduced state vector with $\sigma =1.7\times 10^{-4}$.

$X_{r}$	Bias	$\sigma_{samp}$	$\sigma_{CRLB}$
5000 m	60.40	138.67	133.56
7000 m	88.20	58.42	55.99
2.83 m/s	0.043	0.044	0.044
2.83 m/s	0.037	0.028	0.028

**Table 3 sensors-21-04797-t003:** Performance of the estimator of the plain state vector.

X	Bias	$\sigma_{samp}$	$\sigma_{CRLB}$
5000 m	−44.77	854.72	868.12
7000 m	−68.16	1162.1	1173.60
100 m	7.14	558.55	545.99
2.83 m/s	0.092	1.67	1.65
2.83 m/s	0.194	2.72	2.68

**Table 4 sensors-21-04797-t004:** Performance of the estimator of the reduced state vector.

$X_{r}$	Bias	$\sigma_{samp}$	$\sigma_{CRLB}$
5000 m	65.40	655.61	606.41
7000 m	93.05	831.75	762.56
2.83 m/s	0.034	1.71	1.58
2.83 m/s	0.063	2.76	2.52

**Table 5 sensors-21-04797-t005:** Performance of the estimator of the reduced state vector.

$X_{r}$	Bias	$\sigma_{samp}$	$\sigma_{CRLB}$
5000 m	401.12	281.85	281.17
7000 m	557.24	330.87	319.37
2.83 m/s	0.13	1.58	1.78
2.83 m/s	0.12	1.81	2.02

**Table 6 sensors-21-04797-t006:** Performance of the estimator.

$X_{r}$	Bias	$\sigma_{samp}$	$\sigma_{CRLB}$
5000 m	306.81	219.28	130.08
7000 m	432.46	276.61	115.26
2.83 m/s	0.18	0.74	0.80
2.83 m/s	0.18	0.66	0.71

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
