# Peer review of "TMA from Cosines of Conical Angles Acquired by a Towed Array"

_sensors, 2021, doi:10.3390/s21144797_

Round 1

Reviewer 1 Report

This article presents a solution to the problem of tracking a target following a constant velocity model from a single measurement based on the cosine of the bearing and elevation. The multi-path problem is also considered.

Overall the article presents novel and interesting results, but some aspects of the work would benefit from being better motivated. For instance, the measurement model, although not new, does not seem to be standard, and it would be interesting to understand in which situation it arises. The authors also consider and discuss the cases where the target is "endfire" or "broadside" but later highlight the fact that these events have probability zero (since they require a perfect alignment with the line array). It seems that there are two possibilities: either these events are truly negligible and the discussion surrounding them should be shorter or they remain relevant in practice and this should be explained. For instance, in the case where there is some observation noise, I suspect that targets that are nearly endfire/broadside would be particularly challenging.

It would be good to discuss options in terms of representation of the uncertainty. The authors rely on maximum likelihood estimation which, in practice, means that only a point estimate will be available to the operator. Since the uncertainty could be large in some cases, a notion of confidence would be particularly useful. Giving some directions on this aspect would be sufficient as this is not the focus of the article.

The format is rather unusual in some places, for instance, ll.110-120 mainly consist of one-line paragraphs. The formatting of the inline equations could also be improved, although this might be due to the template used. Several figures and tables are not on the same page as their caption, which should also be avoided.

Other comments:

  • l.42: TMA is defined here but "target motion analysis" is used on l.21
  • l.47: "section III" should be "Section 3"
  • l.76: $\alpha(t)$ is defined via its cosine but not used, except in the appendix
  • l.124: CV is undefined
  • l.137: { m(t) : t \in [0,T] } gives the impression that observations are in continuous time
  • ll.325-326: "However, the minimum standard deviation of the target’s depth is not compatible with the physical constraints." -> this statement would benefit from additional explanations
  • l.327: "section III-A" -> same issue as before
  • l.345: the acronym BOTMA is not defined
  • l.468: "estimable" is defined once more here, it might be better to discuss the concept in more depth the first time it is defined rather than defining it again here

Author Response

We do thank the two reviewers for their thorough reading and their wise suggestions and comments

Hereafter, our replies.

Reply to reviewer #1

This article presents a solution to the problem of tracking a target following a constant velocity model from a single measurement based on the cosine of the bearing and elevation. The multi-path problem is also considered.

Overall the article presents novel and interesting results, but some aspects of the work would benefit from being better motivated. For instance, the measurement model, although not new, does not seem to be standard, and it would be interesting to understand in which situation it arises. The authors also consider and discuss the cases where the target is "endfire" or "broadside" but later highlight the fact that these events have probability zero (since they require a perfect alignment with the line array). It seems that there are two possibilities: either these events are truly negligible and the discussion surrounding them should be shorter or they remain relevant in practice and this should be explained. For instance, in the case where there is some observation noise, I suspect that targets that are nearly endfire/broadside would be particularly challenging.

Reply:

  • The situation we are addressing in this paper is met when the target is in near field; in this case, the two more energetic rays are the direct and the reflected paths (bottom or surface). Most of the cases, the sound is bounced by the sea bottom.
  • The issue of observability requires that all the cases must be studied, even the ones which have a poor probability to occur. This is why, in our analysis, we had to consider these cases.
  • Obviously, when the target has a trajectory close to these special cases, the estimates will have a poor behavior. This situation is encountered in array processing, when a target in endfire is detected.

It would be good to discuss options in terms of representation of the uncertainty. The authors rely on maximum likelihood estimation which, in practice, means that only a point estimate will be available to the operator. Since the uncertainty could be large in some cases, a notion of confidence would be particularly useful. Giving some directions on this aspect would be sufficient as this is not the focus of the article.

Reply: In open literature about TMA, the confidence regions are given by the confidence ellipsoid obtained with the covariance matrix of the estimate. Since the maximum likelihood estimate is asymptotically efficient under nonrestrictive conditions, we use here the Cramer-Rao lower bound to compute such confidence regions.

The format is rather unusual in some places, for instance, ll.110-120 mainly consist of one-line paragraphs. The formatting of the inline equations could also be improved, although this might be due to the template used. Several figures and tables are not on the same page as their caption, which should also be avoided.

Reply: Thank you for this suggestion. We improved the paper by avoiding some useless “carriage returns”.

Other comments:

  • 42: TMA is defined here but "target motion analysis" is used on l.21
  • 47: "section III" should be "Section 3"
  • 76: $\alpha(t)$ is defined via its cosine but not used, except in the appendix
  • 124: CV is undefined
  • 327: "section III-A" -> same issue as before
  • 345: the acronym BOTMA is not defined

Reply: We corrected all these clumsy turns of phrase.  Is $\alpha(t)$ Now $\c_a(t)$ Thanks a lot.

  • 137: { m(t) : t \in [0,T] } gives the impression that observations are in continuous time

Reply: In TMA problems, observability is often analyzed in continuous time (see [15] and [17], for example), even though the noisy measurements are given in discrete time. We added this sentence in the text.

  • 325-326: "However, the minimum standard deviation of the target’s depth is not compatible with the physical constraints." -> this statement would benefit from additional explanations

Reply: We meant that the accuracy of the result are of the same order (or more) that the parameter we estimated. For instance, the standard deviation of the estimated depth is about 550 meters (see Table 3) whereas the actual depth is 100 meters. With such a standard deviation, the estimate of the depth can be negative, that means that the target must be estimated 400 meters up above the sea surface! We gave additive explanation in the text.

  • 468: "estimable" is defined once more here, it might be better to discuss the concept in more depth the first time it is defined rather than defining it again here

Reply: we rephrase this sentence.

Reply to reviewer #2

This papers deals with the observability analysis of a target in constant velocity motion, from measurements of conical angles supplied by a linear array, by considering different known paths of the sound emitted by the source. The discussed problem in this paper is very interesting. However, the following comments should be considered before this manuscript being accepted.

  1. The readability of "problem formulation" needs to be improved, for example, drawing a figure to visualize the geometric relationship about the target of interest, the acoustic and the corresponding coordinates.

Reply: excellent suggestion! We added a missing figure (now Figure 1).

  1. It is better to give an estimate or estimate error curve of the target trajectory directly, due to the target of interest in constant velocity motion instead of being stationary.

Reply: probably our text is confusing. We use a batch procedure (the maximum likelihood estimate computed by using the Gauss Newton routine), hence we cannot display the residue curve along the time (unlike what we should have done if we ran a recursive algorithm, e.g. a EKF). Moreover, in the scenario presented in this paper, the targets are never stationary.

  1. An explanation of knowing the path of the sound emitted by the source should be given, since it is a key problem in multipath target motion analysis.

Reply: The situation we are addressing in this paper is met when the target is in near field; in this case, the two more energetic rays are the direct and the reflected paths (bottom or surface). Most of the cases, the sound is bounced by the sea bottom. In a real situation, the available data must be submitted to a battery of tests to decide what path is the more likely. In this paper, the choice of the direct and reflected paths is a work assumption. We added a sentence in Section 1 to explain it.

Reviewer 2 Report

This papers deals with the observability analysis of a target in constant velocity motion, from measurements of conical angles supplied by a linear array, by considering different known paths of the sound emitted by the source. The discussed problem in this paper is very interesting. However, the following comments should be considered before this manuscript being accepted.

  1. The readability of "problem formulation" needs to be improved, for example, drawing a figure to visualize the geometric relationship about the target of interest, the acoustic and the corresponding coordinates.
  2. It is better to give an estimate or estimate error curve of the target trajectory directly, due to the target of interest in constant velocity motion instead of being stationary.
  3. An explanation of knowing the path of the sound emitted by the source should be given, since it is a key problem in multipath target motion analysis.

Author Response

(The authors gave the same response as above.)

Round 2

Reviewer 1 Report

Most of the comments have been taken into account, but the authors' reply contains the comments:

1) "Obviously, when the target has a trajectory close to these special cases, the estimates will have a poor behavior. This situation is encountered in array processing, when a target in endfire is detected."

and 

2) "In open literature about TMA, the confidence regions are given by the confidence ellipsoid obtained with the covariance matrix of the estimate. Since the maximum likelihood estimate is asymptotically efficient under nonrestrictive conditions, we use here the Cramer-Rao lower bound to compute such confidence regions."

But these remarks have not been added to the manuscript when they could improve its accessibility.

Author Response

We followed your advice and added the two sentences in the text.

A company located in United Kingdom checked English.

Sincerely,

The authors.
